# The Effect of Isosaponarin Derived from Wasabi Leaves on Glutamate Release in Rat Synaptosomes and Its Underlying Mechanism

**DOI:** 10.3390/ijms23158752

**Published:** 2022-08-06

**Authors:** Cheng-Wei Lu, Kun-Chieh Yeh, Kuan-Ming Chiu, Ming-Yi Lee, Tzu-Yu Lin, Su-Jane Wang

**Affiliations:** 1Department of Anesthesiology, Far-Eastern Memorial Hospital, New Taipei City 22060, Taiwan; 2Department of Mechanical Engineering, Yuan Ze University, Taoyuan 32003, Taiwan; 3School of Medicine, Fu Jen Catholic University, New Taipei City 24205, Taiwan; 4Department of Surgery, Taoyuan Armed Forces General Hospital, Taoyuan 325208, Taiwan; 5Department of Surgery, Fu Jen Catholic University Hospital, Fu Jen Catholic University, New Taipei City 24205, Taiwan; 6Division of Cardiovascular Surgery, Cardiovascular Center, Far-Eastern Memorial Hospital, New Taipei City 22060, Taiwan; 7Department of Electrical Engineering, Yuan Ze University, Taoyuan 32003, Taiwan; 8Department of Medical Research, Far-Eastern Memorial Hospital, New Taipei 22060, Taiwan; 9Research Center for Chinese Herbal Medicine, College of Human Ecology, Chang Gung University of Science and Technology, Taoyuan 33303, Taiwan

**Keywords:** isosaponarin, vesicular glutamate release, VGCC, PKC, SNAP-25, MARCKS

## Abstract

Excessive glutamate release is known to be involved in the pathogenesis of neurological diseases, and suppression of glutamate release from nerve terminals is considered to be a treatment strategy. In this study, we investigated whether isosaponarin, a flavone glycoside isolated from wasabi leaves, could affect glutamate release in rat cerebral cortex nerve terminals (synaptosomes). The release of glutamate was evoked by the K^+^ channel blocker 4-aminopyridine (4-AP) and measured by an online enzyme-coupled fluorimetric assay. Isosaponarin produced a concentration-dependent inhibition of 4-AP-evoked glutamate release with a half-maximum inhibition of release value of 22 μM. The inhibition caused by isosaponarin was prevented by eliminating extracellular Ca^2+^ or by using bafilomycin A1, an inhibitor of synaptic vesicle exocytosis. Isosaponarin decreased intrasynaptosomal rises in Ca^2+^ levels that were induced by 4-AP, without affecting the synaptosomal membrane potential. The isosaponarin-induced inhibition of glutamate release was significantly prevented in synaptosomes that were pretreated with a combination of the calcium channel blockers ω-conotoxin GVIA (N-type) and ω-agatoxin IVA (P/Q-types). The protein kinase C (PKC) pan-inhibitor GF109203X and the Ca^2+^-dependent PKC inhibitor Go6976 abolished the inhibition of glutamate release by isosaponarin, while the Ca^2+^-independent PKC inhibitor rottlerin did not show any effect. The results from immunoblotting assays also showed that isosaponarin lowered PKC, PKCα, synaptosomal-associated protein of 25 kDa (SNAP-25), and myristoylated alanine-rich C-kinase substrate (MARCKS) phosphorylation induced by 4-AP. In addition, FM1-43-labeled synaptic vesicles in synaptosomes showed that treatment with isosaponarin resulted in an attenuation of the 4-AP-induced decrease in fluorescence intensity that is consistent with glutamate release. Transmission electron microscopy of synaptosomes also provided evidence that isosaponarin altered the number of synaptic vesicles. These results indicate that isosaponarin suppresses the Ca^2+^-dependent PKC/SNAP-25 and MARCKS pathways in synaptosomes, causing a decrease in the number of available synaptic vesicles, which inhibits vesicular glutamate release from synaptosomes.

## 1. Introduction

Glutamate is the major excitatory neurotransmitter in the central nervous system (CNS); however, excess levels may cause significant oxidative glutamate toxicity and nerve cell injury, which are closely related to the pathogenesis of many CNS diseases, such as ischemic stroke, epilepsy, and neurodegenerative diseases [1,2]. Therefore, the level of glutamate must be strictly maintained within the nervous system, and agents that modulate its release have potential as therapeutics [3,4,5]. Notably, the use of natural plants in the treatment of neurological diseases has increased considerably [6,7], and many plant-derived compounds are effective in inhibiting presynaptic glutamate release [8,9,10]. Wasabi (*Wasabia japonica*), a perennial plant that is native to Japan and belongs to the Brassicaceae family, is commonly used in food as a pungent spice [11]. Wasabi has numerous health-promoting effects, such as antioxidant, anti-inflammatory, anti-obesity, anti-cancer, antiviral, and neuroprotective activities [12,13,14,15,16]. Previous studies have shown that wasabi leaves contain flavonoids [11], phenylpropanoids [17], terpenoids [18], and carotenoids [18]. In addition, functional studies have demonstrated biological activity, especially antioxidative effects [17]. Numerous flavonoids with antioxidative activity are known to reduce glutamate release and counteract glutamate-induced oxidative damage to neurons [19,20,21]. In the present study, isosaponarin, a 4′-*O*-glucosyl-6-*C*-glucosyl apigenin (Figure 1A), was chosen because it is one of the flavonoids found in wasabi leaves [11,22]; however, its role in the regulation of glutamate release has not yet been clarified. Therefore, in this study we used isolated nerve terminals (synaptosomes) prepared from the rat cortex to evaluate the effect of isosaponarin on glutamate release, and clarified the related mechanisms that contribute to its effect. Synaptosomes are isolated nerve endings that carry the structural features of neuronal terminals in vivo and are, therefore, extensively used as an in vitro model to evaluate presynaptic effects on neurotransmitter release [23].

## 2. Results

### 2.1. Isosaponarin Reduces 4-AP-Triggered Ca^2+^-Dependent Vesicular Glutamate Release from Synaptosomes

In order to examine the effect of isosaponarin on glutamate release, isolated nerve terminals from the rat cerebral cortex were depolarized with the K^+^-channel blocker 4-AP. 4-AP destabilizes the membrane potential, and is thought to cause repetitive spontaneous Na^+^-channel-dependent depolarization that closely approximates in vivo depolarization of the synaptic terminal, which leads to the activation of VGCCs and neurotransmitter release [24]. Under control conditions, 4-AP (1 mM) triggered glutamate release of 7.7 ± 0.1 nmol/mg/5 min from synaptosomes that were incubated in the presence of 1.2 mM CaCl_2_. Preincubation with synaptosomes with isosaponarin (30 μM) for 10 min before 4-AP addition did not alter basal glutamate release but significantly decreased the release of glutamate that was triggered by 4-AP to 4.0 ± 0.3 nmol/mg/5 min [t(15) = 42.9, *p* < 0.001; Figure 1B]. The inhibition produced by isosaponarin was concentration dependent, with 20, 30, 50, and 100 μM isosaponarin decreasing glutamate release by 23 ± 2%, 50 ± 2%, 61 ± 1%, and 76 ± 1%, respectively (*p* < 0.001; Figure 1C). In addition, the release of glutamate triggered by 4-AP was significantly reduced in the absence of external Ca^2+^ [F(2,12) = 9077.9, *p* < 0.001]. However, this Ca^2+^-independent component of 4-AP-triggered glutamate release was not significantly affected by isosaponarin (*p* = 0.98; Figure 1C). Similar results were also obtained using bafilomycin A1, a vacuolar H^+^-ATPase inhibitor that inhibits neurotransmitter uptake into synaptic vesicles and subsequent synaptic vesicle exocytosis [25]. As shown in Figure 1C, bafilomycin A1 (0.1 μM) significantly decreased the release of glutamate triggered by 4-AP [F(2,12) = 5315.8, *p* < 0.001]. In the presence of bafilomycin A1, isosaponarin produced no significant inhibition of 4-AP-triggered glutamate release (*p* = 0.99).

### 2.2. Suppression of N-Type and P/Q-Type Ca^2+^ Channels Mediates Isosaponarin-Induced Inhibition of Vesicular Glutamate Release

In Figure 2A, the effect of isosaponarin on 4-AP-evoked increases in [Ca^2+^]_C_ was assessed using Fura-2. Elevation in [Ca^2+^]_C_ occurred after the addition of 4-AP (1 mM) to synaptosomes, and reached a plateau level of 178.8 ± 1.2 nM. Preincubation with isosaponarin (30 μM) before 4-AP addition significantly reduced the rise in [Ca^2+^]_C_ evoked by 4-AP [t(8) = 6.1, *p* < 0.001], and the plateau level in the presence of isosaponarin was 143.6 ± 4.1 nM. Figure 2B shows the effects of different VGCC blockers, known to act selectively on the N-type (ω-conotoxin GVIA) or on the P/Q-type (ω-agatoxin IVA) Ca^2+^ channels [26], on the isosaponarin-mediated inhibition of glutamate release. ω-conotoxin GVIA (2 μM) or ω-agatoxin IVA (0.5 μM) significantly decreased the release of glutamate evoked by 4-AP [ω-conotoxin GVIA, F(2,12) = 273.1, *p* < 0.001; ω-agatoxin IVA, F(2,12) = 265.2, *p* < 0.001]. In the presence of ω-conotoxin GVIA (2 μM) alone or ω-agatoxin IVA (0.5 μM) alone, isosaponarin caused a significant inhibition of 4-AP-evoked glutamate release (*p* < 0.001). However, in the presence of both ω-conotoxin GVIA (2 μM) and ω-agatoxin IVA (0.5 μM), isosaponarin produced no significant inhibition of 4-AP-evoked glutamate release (*p* = 1).

### 2.3. Isosaponarin Does Not Change the Synaptosomal Membrane Potential

Since neurotransmitter release can be modulated by regulating the plasma membrane potential and consequently altering the Ca^2+^ influx [27], we used the membrane potential-sensitive dye DiSC_3_(5) to determine whether isosaponarin affects the synaptosomal membrane potential. The dye became incorporated into the synaptosomal plasma membrane lipid bilayer. Upon depolarization with 4-AP, the release of the dye from the membrane bilayer is indicated as an increase in fluorescence [28]. As shown in Figure 3, 4-AP addition caused an increase in DiSC_3_(5) fluorescence. Preincubation with isosaponarin (30 μM) for 10 min before 4-AP addition did not alter either the basal DiSC_3_(5) fluorescence or the rise in DiSC_3_(5) fluorescence evoked by 4-AP [t(8) = −0.5, *p* = 0.7].

### 2.4. Isosaponarin-Induced Inhibition of Vesicular Glutamate Release Is Prevented by Ca^2+^-Dependent PKC Inhibition

PKC is a well-known signaling molecule involved in the regulation of glutamate exocytosis [29]. In order to explore the possibility that the isosaponarin-mediated inhibition of glutamate release involves PKC action in our model system, we assessed glutamate release in the presence or absence of isosaponarin in synaptosomes in which the activity of PKC was blocked. As shown in Figure 4A, the PKC pan-inhibitor bisindolylmaleimide I (GF109203X) (10 μM) significantly decreased the release of glutamate evoked by 4-AP [F(2,12) = 1423.2, *p* < 0.001]. In the presence of GF109203X, isosaponarin (30 μM) induced a statistically insignificant inhibition of 4-AP-evoked glutamate release compared with the inhibition produced by isosaponarin alone. The addition of 1 μM Go6976 [12-(2-cyanoethyl)-6,7,12,13-tetrahydro-13-methyl-5-oxo-5H-indolo(2,3-a)pyrrolo(3,4-c)-carbazole], an inhibitor of Ca^2+^-dependent PKCs (α, βI, βII, and γ), also significantly decreased 4-AP-evoked glutamate release [F(2,12) = 928.3, *p* < 0.001]. After treatment with Go6976, isosaponarin failed to produce any significant effect on glutamate release *(p* = 0.98). Rottlerin (1 μM), an inhibitor of Ca^2+^-independent PKCs (δ, ε, η, and θ), had significant effect on 4-AP-evoked glutamate release [F(2,12) = 3602.2, *p* < 0.001]. In the presence of rottlerin, however, the isosaponarin-induced inhibition of glutamate release was not affected (*p* < 0.001). In addition, we used Western blotting to examine the effect of isosaponarin on the phosphorylation of PKC and PKCα, a Ca^2+^-dependent PKC isozyme that is highly expressed in synaptosomes [30]. Figure 4B shows that the phosphorylation levels of PKC and PKCα were significantly increased in synaptosomes treated with 4-AP (1 mM) in the presence of 1.2 mM CaCl_2_ [pPKC, F(2,12) = 233.3, *p* < 0.001; pPKCα, F(2,12) = 160.4, *p* < 0.001]. Preincubation with isosaponarin (30 μM) for 10 min before 4-AP addition significantly decreased the 4-AP-enhanced phosphorylation of PKC and PKCα (*p* < 0.001). 4-AP and isosaponarin did not affect the protein expression levels of PKC and PKCα (*p* = 0.9).

### 2.5. Isosaponarin Decreases SNAP-25 and MARCKS Phosphorylation Evoked by 4-AP in Synaptosomes

PKC has multiple targets within the exocytotic machinery, including SNAP-25 and MARCKS, which are thought to interfere with synaptic vesicle function [29,31]. We next investigated whether isosaponarin can alter the phosphorylation levels of SNAP-25 and MARCKS. As shown in Figure 5, 4-AP (1 mM) significantly increased the phosphorylation levels of SNAP-25 at serine187 (Ser187) and MARCKS at serine152/156 (Ser152/156) [pSNAP-25 (Ser187), F(2,12) = 10.3, *p* < 0.01; pMARCKS (Ser152/156), F(2,12) = 68.5, *p* < 0.001]. In the presence of isosaponarin (30 μM), the 4-AP-enhanced phosphorylation of SNAP-25 (Ser187) was significantly reduced (*p* < 0.05). 4-AP and isosaponarin did not affect the protein expression levels of SNAP-25 (Ser187) (*p* = 0.9). The results suggest that the isosaponarin-mediated inhibition of 4-AP-evoked vesicular glutamate exocytosis is related to the inhibition of SNAP-25 and MARCKS phosphorylation.

### 2.6. Isosaponarin Decreases the Release of FM1-43 from Synaptosomes

In order to confirm that isosaponarin affects synaptic vesicle release during 4-AP stimulation, we performed an experiment measuring the release of FM1-43, which tracks actively recycling synaptic vesicles [32]. FM1-43 is incorporated into synaptic vesicles during endocytosis. Accumulated FM1-43 is released during exocytosis when the lumen of the synaptic vesicle is exposed to the extracellular medium. The release of FM1-43 from synaptic vesicles is detected as a decrease in fluorescence [33]. As shown in Figure 6A, 4-AP (1 mM) caused a decrease in FM1-43 fluorescence in the presence of CaCl_2_. When synaptosomes were preincubated with isosaponarin (30 μM) for 10 min before 4-AP addition, the decrease in fluorescence was inhibited, which was significantly different from the loss of fluorescence detected under control conditions [t(8) = −9.3, *p* < 0.001]. Figure 6B shows the relative fluorescence of FM1-43, where the fluorescence intensity evoked by 4-AP was attenuated in the 30-micromolar isosaponarin-treated synaptosomes.

### 2.7. Isosaponarin Inhibits the 4-AP-Induced Decrease in the Number of Synaptic Vesicles in Synaptosomes

In Figure 7A, the synaptosome ultrastructure and the number of synaptic vesicles were observed with transmission electron microscopy (TEM). In the control group, normal synaptosome ultrastructure was observed: a clear synaptic cleft, postsynaptic density (PSD), and certain numbers of synaptic vesicles in the presynaptic membrane. In 4-AP-treated synaptosomes, the number of synaptic vesicles decreased. However, in the isosaponarin-pretreated synaptosomes, the 4-AP-caused synaptic vesicle reduction was attenuated. Figure 7B shows significant differences in the numbers of synaptic vesicles between control conditions, following treatment with 4-AP, or after treatment with isosaponarin and 4-AP (*p* < 0.05).

## 3. Discussion

Excess glutamate exocytosis from nerve terminals alters synaptic plasticity, contributing to the pathogenesis of most neurological diseases [1,2]. In this context, drugs that inhibit synaptic glutamate release could be therapeutic for disease progression, and natural compounds represent potential candidates for such a therapeutic approach [8,9,10]. Here, we studied the effect of isosaponarin, a flavone glycoside natural product derived from wasabi leaf, on glutamate release in rat cortex nerve terminals (synaptosomes).

### 3.1. Isosaponarin Decreases 4-AP-Evoked Vesicular Glutamate Release by Blocking N- and P/Q-Type Ca^2+^ Channels

We observed that isosaponarin inhibited 4-AP-triggered glutamate release from synaptosomes. The release of glutamate triggered by 4-AP from neurons is mediated by Ca^2+^-dependent vesicular exocytosis and reverse transport by glutamate transporters [34,35]. In this study, the inhibitory effect of isosaponarin on 4-AP-triggered glutamate release was not observed in extracellular Ca^2+^-free solution. The fraction of glutamate release by 4-AP in the absence of external Ca^2+^ is completely dependent on the entrance of Na^+^ into the nerve ending, and involves reversal of the glutamate transporters [34]. Therefore, the involvement of decreased reverse transport through glutamate transporters in the isosaponarin-mediated inhibition of 4-AP-triggered glutamate release from synaptosomes is not included in our observation. In addition, the isosaponarin effect was completely prevented by the vesicular transport inhibitor bafilomycin A1. This finding supports the conclusion that isosaponarin inhibits 4-AP-evoked Ca^2+^-dependent vesicular glutamate release from synaptosomes. In addition, an intracellular Ca^2+^ rise in nerve terminals, mainly mediated via N- and P/Q-type Ca^2+^ channels, is crucial for efficient vesicular neurotransmitter release [26,36,37]. Here, we found that an elevation of [Ca^2+^]_C_ resulting from nerve terminal depolarization with 4-AP was reduced by isosaponarin. Furthermore, although isosaponarin-induced inhibition of glutamate release persisted after an individual blockade of N- or P/Q-type Ca^2+^ channels, a combined blockade of both channel types prevented the effect of isosaponarin. These results suggest that the inhibition of N- and P/Q-type Ca^2+^ channel activities together potentially underlies the influence of isosaponarin on vesicular glutamate release from nerve terminals. In addition, neurotransmitter release can be modulated by regulating the plasma membrane potential, which consequently alters Ca^2+^ influx [27]. However, the suppression of VGCCs by isosaponarin is not because of an indirect effect through modulation of membrane potential changes, and hence synaptosomal excitability, as we observed that 4-AP-evoked membrane potential depolarization measured with a membrane-potential sensitive dye DiSC_3_(5) was unaffected by isosaponarin. Whether the inhibition of VGCCs by isosaponarin reflects a direct interaction with VGCCs requires further investigation.

### 3.2. Suppression of the Ca^2+^-Dependent PKC/SNAP-25 and MARCKS Pathways and the Consequent Reduction in Available Synaptic Vesicles May Account for Isosaponarin-Induced Inhibition of Vesicular Glutamate Release

PKC, a presynaptically enriched multifunctional enzyme, is known to be a crucial modulator of the exocytotic pathway, where it is involved in the enhancement of both the priming and fusion steps of vesicle exocytosis through phosphorylation of several proteins of the exocytotic machinery [38,39]. The phosphorylation targets of PKC include SNAP-25 and MARCKS [29,31]. SNAP-25 is a component of the SNARE complex, which together with syntaxin-1 and synaptobrevin, mediates synaptic vesicle fusion with the plasma membrane. SNAP-25 is phosphorylated by PKC at Ser187, which is known to enhance Ca^2+^-dependent release by regulating the refilling of synaptic vesicles [40,41]. MARCKS, another prominent substrate of PKC, regulates F-actin dynamics. Phosphorylation of MARCKS by PKC at Ser152/156 results in F-actin disassembly and increases vesicle recruitment to the plasma membrane [42,43,44]. Therefore, decreased phosphorylation of SNAP-25 or MARCKS by PKC reduces the number of available synaptic vesicles as well as their release probability [3,4]. Based on our finding that isosaponarin inhibits glutamate release through a mechanism supported by N- and P/Q-type Ca^2+^ channels, we infer that the decreased Ca^2+^ level upon isosaponarin treatment has an impact on Ca^2+^-dependent PKC activity, and consequently on PKC-mediated synaptic vesicle fusion. Several lines of evidence support this hypothesis. Firstly, in synaptosomes treated with the Ca^2+^-dependent PKC inhibitor Go6976, the inhibitory effect of isosaponarin on 4-AP-evoked glutamate release was completely blocked. However, the inhibition of release by isosaponarin was unaffected in the presence of rottlerin, an inhibitor of Ca^2+^-independent PKC. Secondly, isosaponarin reduced the 4-AP-induced phosphorylation levels of PKC, SNAP-25, and MARCKS. Thirdly, in the FM1-43 analysis, the observed decay in FM1-43 fluorescence is a measure of activity-dependent synaptic vesicular uptake and release for neurons [32,45]. We found that isosaponarin markedly reduces the release of FM1-43 from synaptosomes during 4-AP stimulation, which indicates that the delivery of synaptic vesicles to the active zone is inhibited. Fourthly, using TEM, we also observed that the 4-AP-caused synaptic vesicle reduction was markedly inhibited in isosaponarin-treated synaptosomes. These results indicate that isosaponarin inhibits 4-AP-evoked glutamate release from synaptosomes by reducing the number of available synaptic vesicles through Ca^2+^-dependent PKC/SNAP-25 and MARCKS mechanisms. In addition to SNAP-25 and MARCKS, however, the possible involvement of other molecules should be considered. For example, Munc13-1 and Munc13-8 have been reported to be phosphorylated by PKC and involved in the regulation of synaptic vesicle recycling [46,47]. Therefore, reduced phosphorylation levels of Munc13-1 and Munc13-8 in isosaponarin-treated synaptosomes may also contribute to the reductions in available synaptic vesicles and release probability.

In the present study, isosaponarin inhibited 4-AP-evoked vesicular glutamate release from rat cortical nerve terminals at concentrations ranging from 20 to 100 μM. Although the dose of isosaponarin used in our study to produce the effect was higher, the action of isosaponarin was specific. For example, isosaponarin reduced the 4-AP-induced increase in [Ca^2+^]_C_, whereas it did not affect 4-AP-mediated membrane potential depolarization. Furthermore, the isosaponarin-mediated inhibition of glutamate release was significantly blocked by Ca^2+^-dependent but not Ca^2+^-independent PKC inhibitors. In addition, although we demonstrated that isosaponarin has an inhibitory effect on glutamate release from nerve terminals, the relevance of our findings to in vivo glutamate homeostasis and glutamate excitotoxicity requires further exploration.

Taken together, we report that the isosaponarin-mediated decrease in 4-AP-evoked glutamate release may be explained through the inhibition of N- and P/Q-type Ca^2+^ channels, which interrupts SNAP-25 and MARCKS phosphorylation by PKC, and leads to a subsequent reduction in available synaptic vesicles in nerve terminals (Figure 8). These data shed light on the mechanism of action of isosaponarin in the brain, which could have implications for the role of isosaponarin in neurological diseases associated with glutamate excitotoxicity.

## 4. Materials and Methods

### 4.1. Animals

Rats (male, 150−200 g) were obtained from BioLASCO (Taipei, Taiwan), and were housed in the animal center of Fu Jen Catholic University under environmentally controlled conditions (temperature = 21−25 °C, humidity = 40%) on a 12-hour light/dark cycle with food and water *ad libitum*. All the rats were euthanized by cervical dislocation, followed by decapitation, and the cortices were rapidly removed. The experimental procedures were approved by the Animal Ethics Committee of Fu Jen Catholic University (A11113), according to the Guidelines for Animal Care and Use of the National Institutes of Health. In line with the 3Rs rules (replacement, refinement, and reduction), every effort was made to reduce the number of animals required to obtain statistically reliable results. The total number of rats used in the study was 34; specifically, the measurements of glutamate release, Ca^2+^ concentrations, membrane potential, and protein phosphorylation in nerve terminals—20 animals; FM1-43 release and image—5 animals; TEM—9 animals.

### 4.2. Isolation of Nerve Terminals (Synaptosomes) from the Cortex Regions of the Rat Brains

Rat cortical purified synaptosomes were prepared as previously described [23]. Briefly, the rat cerebral cortex was homogenized in 0.32 M sucrose (pH 7.4) and centrifuged 10 min at 3000× *g*. The supernatant was recovered and centrifuged again for 15 min at 14,000× *g*. After discarding the supernatant, the pellet was resuspended in HEPES buffered medium (mM: NaCl 128, KCl 2.4, MgSO_4_ 1.2, KH_2_PO_4_ 1.2, CaCl_2_ 1.0, HEPES 10, and glucose 10, pH 7.4) and then gently layered on a discontinuous Percoll gradient (3, 10, and 23% Percoll in Tris-buffered sucrose). After centrifugation at 32,500× *g* for 7 min, the layer between 10 and 23% Percoll (synaptosomal fraction) was collected in HEPES buffered medium.

### 4.3. Glutamate Release Assay

Glutamate release from synaptosomes was detected through a glutamate dehydrogenase reaction, as described previously [48,49]. Briefly, synaptosomes (0.5 mg/mL of final protein concentration) were incubated in HEPES-buffered medium containing glutamate dehydrogenase (20 U/mL) (Sigma, St. Louis, MO, USA), β-nicotinamide adenine dinucleotide (NAD^+^, 1 mM), and CaCl_2_ (1.2 mM), at 37 °C for 5 min. In the presence of glutamate, glutamate dehydrogenase reduced NAD^+^ to NADH, a product that fluoresces (excitation and emission wavelengths of 340 and 460 nm, respectively). Fluorescence intensity of NADH was measured in a stirred thermostated cuvette (37 °C) using a PerkinElmer LS55 spectrofluorimeter. Endogenous glutamate released from the synaptosomes to the incubation medium was detected as an increase in NADH fluorescence. Released glutamate was calibrated by a standard of exogenous glutamate (5 nmol) and expressed as nanomoles of glutamate per milligram of synaptosomal protein (nmol/mg protein). Values quoted in the text and depicted in bar graphs represent the levels of glutamate that were cumulatively released after 5 min of depolarization, and are expressed as nmol/mg protein/5 min.

### 4.4. Cytosolic Ca^2+^ Measurements Using Fura-2

As described previously [49,50], synaptosomes were resuspended in HEPES-buffered medium containing Fura-2-acetoxymethyl ester (5 μM) and CaCl_2_ (0.1 mM) for 30 min at 37 °C. The synaptosome suspension was centrifuged (3000× *g* for 1 min), and the pellet was resuspended in HEPES-buffered medium. CaCl_2_ (1.2 mM) was added, and Fura-2/Ca fluorescence was measured in a Perkin-Elmer LS55 spectrofluorimeter at excitation wavelengths of 340 and 380 nm (emission wavelength 505 nm). Cytosolic Ca^2+^ concentration ([Ca^2+^]_C_, nM) was calculated by using calibration procedures and equations described previously [51,52].

### 4.5. Membrane Potential Measurement Using 3,3,3-Diethylthiacarbocyanine Iodide (DiSC_3_(5))

As described previously [28,49], synaptosomes were re-suspended in HEPES-buffered medium containing DiSC_3_(5) (5 μM) and CaCl_2_ (1.2 mM) for 10 min at 37 °C. DiSC_3_(5) fluorescence was measured in a Perkin-Elmer LS55 spectrofluorimeter at excitation and emission wavelengths of 646 nm and 674 nm, respectively. Results are expressed in fluorescence units.

### 4.6. Western Blot

Western blotting was performed on synaptosomes, as described previously [50]. Equal amounts of lysate protein were run on sodium dodecyl sulfate polyacrylamide gel electrophoresis (SDS-PAGE) and then electrophoretically transferred to polyvinylidene difluoride membrane (GE Healthcare UK Ltd., Amersham, UK). The membrane was first blocked with TBST buffer (500 mM NaCl, 20 mM Tris-HCl [pH7.4], and 0.1% Tween-20) containing 5% non-fat dry milk, and then incubated with primary antibody overnight at 4 °C and horseradish peroxidase-conjugated secondary antibody (1:2000, Gentex, Zeeland, MI, USA) for another 1 h. Bound antibodies were detected with an enhanced chemiluminescence (ECL) system. Quantification of the signals was obtained using Image software. From the obtained intensity for each band the background subtraction was made, and all the values were normalized to β-actin. The primary antibodies used were PKC (1:600, Abcam, Cambridge, UK); pPKC (1:1000; Cell Signaling, Beverly, MA, USA); PKCα (1:600; Cell Signaling, Beverly, MA, USA); pPKCα (1:2000; Abcam, Cambridge, UK); SNAP-25 (1:20,000; Abcam, Cambridge, UK); pSNAP-25 (Ser187) (1:2000; Abcam, Cambridge, UK); pMARCKS (Ser152/156) (1:250; Cell Signaling, Beverly, MA, USA); and β-actin (1:1000; Cell Signaling, Beverly, MA, USA).

### 4.7. FM1-43 Release and Image Assay

Synaptic vesicle fusion with the plasma membrane was measured via the release of the fluorescent dye FM1-43, as described previously [33]. In brief, synaptosomes were incubated in HEPES-buffered medium containing CaCl_2_ (1.2 mM) and FM1-43 (100 μM) at 37 °C in a stirred test tube. After 3 min of stimulation with KCl (50 mM) to load FM1-43, synaptosomes were washed with HEPES-buffered medium and centrifuged (3000× *g* for 1 min). The pellet was then resuspended in HEPES-buffered medium. CaCl_2_ (1.2 mM) was added, and the release of accumulated FM1-43 was induced by 4-AP (1 mM) addition. FM1-43 fluorescence was measured as the decrease in fluorescence upon release of the dye into solution (excitation 488 nm, emission 540 nm) in a Perkin-Elmer LS55 spectrofluorimeter. Data were presented as the FM1-43 fluorescence.

Regarding FM1-43 image assaying, preloaded FM1-43 synaptosomes were placed on coverslips (diameter 20 mm) that were pre-coated with poly-L-lysine for 40 min at 4 °C. After rinsing with HEPES-buffered medium 3 times, the coverslips were incubated with HBM-Ca^2+^ solution that contained 1 mM of 4-AP for 10 min. The coverslips were then washed 3 times with HEPES-buffered medium, and coverslipped with fluorescence mounting medium (DAKO North America, Inc., Carpinteria, CA, USA). Images were observed at a magnification of 400×, using upright fluorescence microscopy (LeicaDM2000 LED, Wetzlar, Germany). Images were captured using a CCD camera (SPOT RT3, Diagnostic Instruments, Sterling Heights, MI, USA).

### 4.8. The Synaptosomes TEM

Synaptosomes were fixed in 4% paraformaldehyde and 0.1% glutaraldehyde, and were stored in a refrigerator at 4 °C overnight. The next day, synaptosomes were washed with 0.1 M phosphate buffer, postfixed in 1% osmium tetroxide for 2 h, dehydrated, and embedded in epoxy resin. Subsequently, synaptosomes were cut at 70 nm by using an ultramicrotome (EM UC7, Leica Microsystems, Wetzlar, Germany). Thin sections were counter-stained with uranyl acetate and lead citrate, and then examined in a TEM (JEM-1400, JEOL, Tokyo, Japan).

### 4.9. Statistical Analysis

Results were expressed as mean ± S.E.M. Statistical analyses were carried out using *t*-tests and analyses of variance (ANOVA). Differences were considered significant when *p* ≤ 0.05 (GraphPad Prism 8, version 8.4.3 Software, GraphPad Software, San Diego, CA, USA).

### 4.10. Chemicals

Isosaponarin (purity > 98%) was purchased from ChemFace (Wuhan, China). The agents 4-AP, bafilomycin A1, GF109203X, Go6976, and rottlerin were purchased by Tocris Bioscience (Bristol, UK). ω-conotoxin GVIA and ω-agatoxin IVA were purchased from Alomone labs (Jerusalem, Israel). Isosaponarin dissolved in 0.1% dimethylsulfoxide (DMSO) was added 10 min before 4-AP addition, and other drugs (bafilomycin A1, ω-conotoxin GVIA, ω-agatoxin IVA, GF109203X, Go6976, and rottlerin) were added 10–20 min before this.

## Figures and Tables

**Figure 1 ijms-23-08752-f001:**
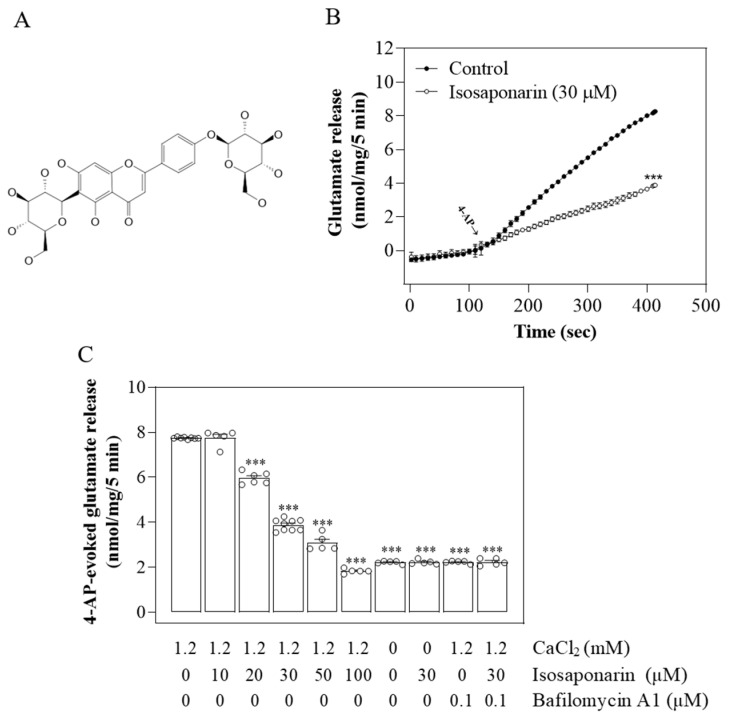
Inhibitory effect of isosaponarin on the 4-AP-evoked glutamate release in rat cerebrocortical nerve terminals. (**A**) The chemical structure of isosaponarin. (**B**) Preincubation with synaptosomes with isosaponarin for 10 min before 4-AP addition significantly inhibited the release of glutamate evoked by 4-AP (1 mM). (**C**) Inhibition of 4-AP-evoked glutamate release by isosaponarin in a dose-dependent manner; this inhibition was prevented in the absence of extracellular Ca^2+^ or in the presence of the vesicular transporter inhibitor bafilomycin A1. Isosaponarin was added 10 min before depolarization, and bafilomycin A1 was added 10 min before this. Data are presented as mean ± SEM (n = 5–14 per group). *** *p* < 0.001 vs. control group.

**Figure 2 ijms-23-08752-f002:**
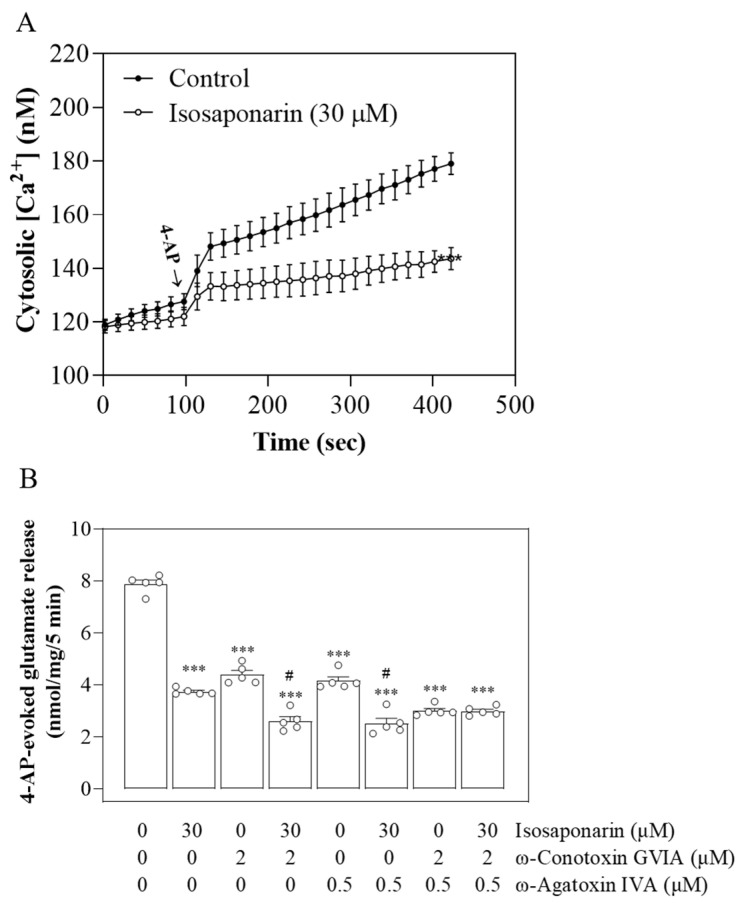
Isosaponarin inhibits the [Ca^2+^]_C_ and the N- and P/Q-type Ca^2+^ channel-mediated glutamate release. (**A**) [Ca^2+^]_C_ was monitored using Fura-2. Synaptosomes were stimulated with 4-AP (1 mM) in the absence (control) or presence of isosaponarin that was added 10 min before stimulation. (**B**) Effect of isosaponarin on 4-AP-evoked glutamate release in the presence of the Ca^2+^ channel toxins ω-conotoxin GVIA or ω-agatoxin IVA, which was added either alone or in combination. Data are presented as mean ± SEM (n = 5 per group). *** *p* < 0.001 vs. control group; # *p* < 0.001 vs. ω-conotoxin GVIA- or ω-agatoxin IVA-treated group.

**Figure 3 ijms-23-08752-f003:**
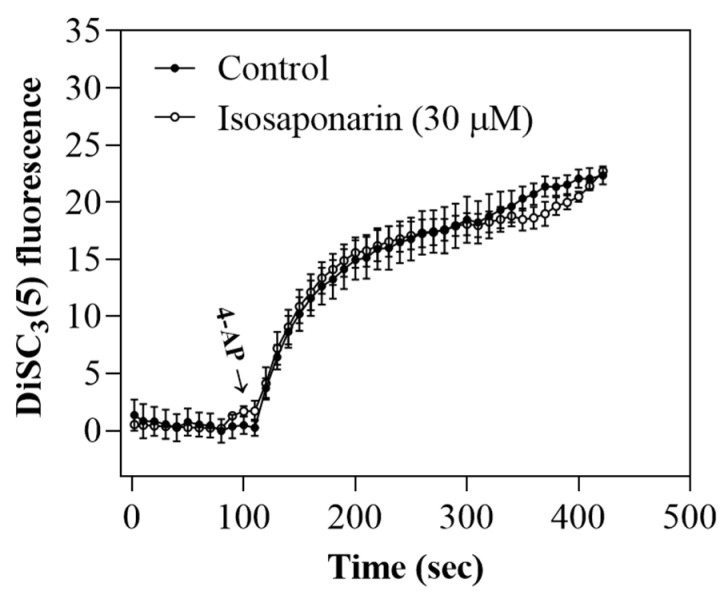
Isosaponarin fails to alter the synaptosomal membrane potential. Synaptosomal membrane potential was measured using DiSC_3_(5) in the absence (control) and in the presence of isosaponarin, added 10 min before stimulation with 1 mM 4-AP. Data are presented as mean ± SEM (n = 5 per group).

**Figure 4 ijms-23-08752-f004:**
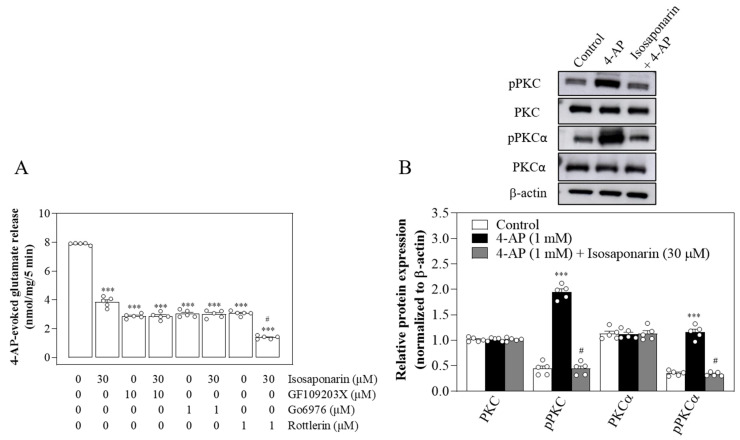
Inhibition of PKC is involved in the inhibition of 4-AP-evoked glutamate release by isosaponarin. (**A**) Effect of isosaponarin on 4-AP-evoked glutamate release in the presence of the PKC inhibitors GF109203X, Go6976, or rottlerin. (**B**) Effect of isosaponarin on the phosphorylation and protein levels of PKC and PKCα. Upper panel, Western blot bands of PKC, pPKC, PKCα, pPKCα, and β-actin. Lower panel, expressions of PKC, pPKC, PKCα, and pPKCα were normalized with β-actin. Isosaponarin was added 10 min before the addition of 4-AP, and other drugs were added 20 min before this. Data are presented as mean ± SEM (n = 5 per group). *** *p* < 0.001 vs. control group; # *p* < 0.001 vs. rottlerin- or 4-AP-treated group.

**Figure 5 ijms-23-08752-f005:**
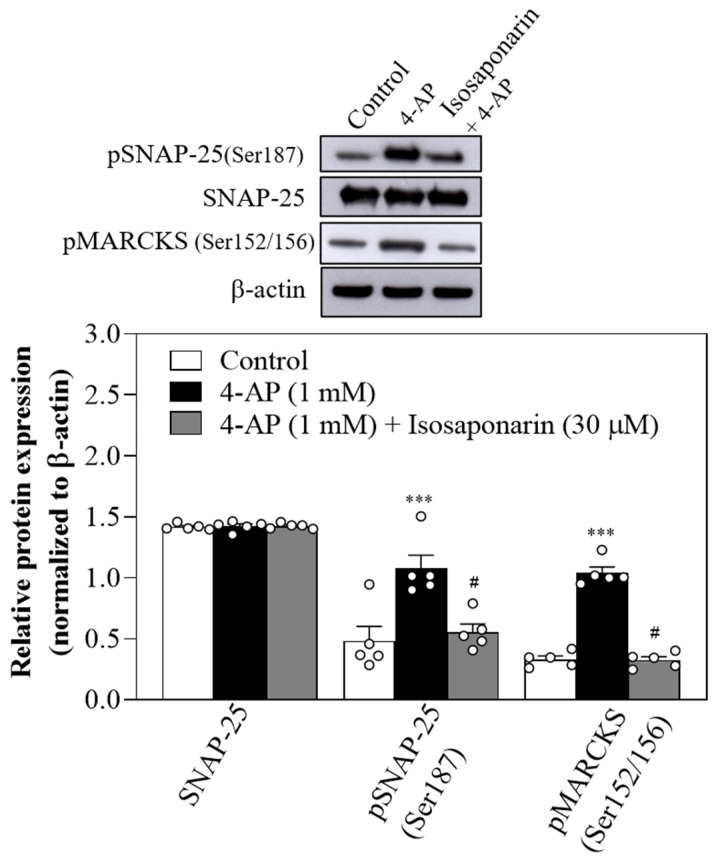
Isosaponarin decreases 4-AP-evoked phosphorylation of SNAP-25 and MARCKS. Upper panel, Western blot bands of SNAP-25, pSNAP-25 (Ser187), pMARCKS (Ser152/156), and β-actin. Lower panel, expressions of SNAP-25, pSNAP-25 (Ser187), and pMARCKS were normalized with β-actin. Isosaponarin was added 10 min before the addition of 4-AP. Data are presented as mean ± SEM (n = 5 per group). *** *p* < 0.001 vs. control group; # *p* < 0.001 vs. 4-AP-treated group.

**Figure 6 ijms-23-08752-f006:**
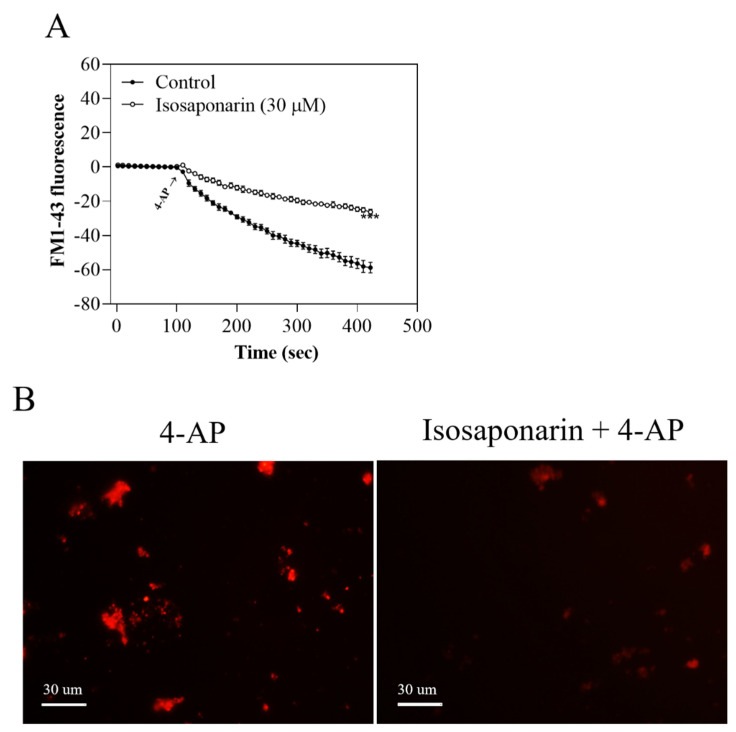
Isosaponarin decreases the release and fluorescence intensity of FM1-43 evoked by 4-AP in synaptosomes. (**A**) FM1-43 release was evoked by 4-AP in the absence (control) or presence of isosaponarin that was added 10 min before the addition of 4-AP. Images were observed at a magnification of 100×. Scale bar, 30 mm. (**B**) FM1-43 images in synaptosomes treated with 4-AP, or with isosaponarin and 4-AP. Data are presented as mean ± SEM (n = 5 per group). *** *p* < 0.001 vs. control group.

**Figure 7 ijms-23-08752-f007:**
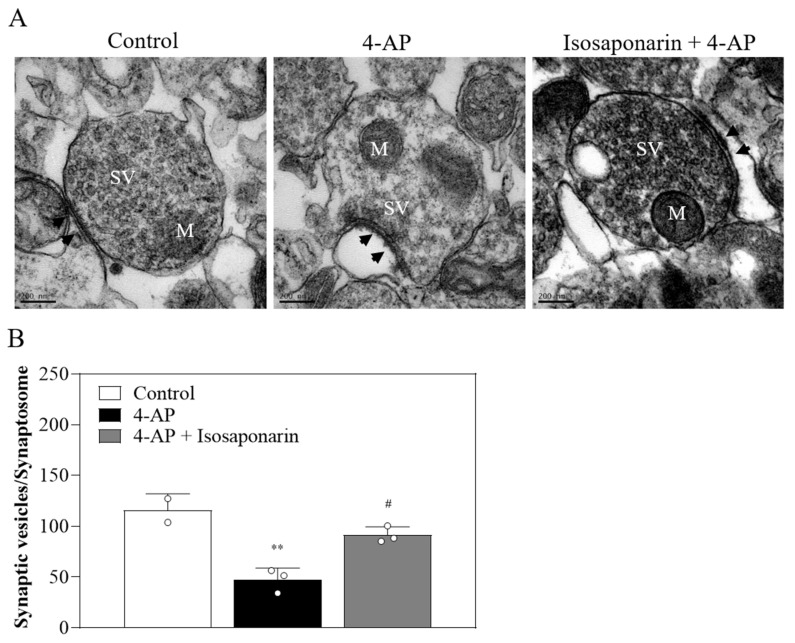
Transmission electron microscopy of synapse ultrastructure (**A**) and numbers of synaptic vesicles (**B**) Each synaptosome contains mitochondria (M), numerous synaptic vesicles (SV), synaptic cleft, and postsynaptic density (arrowhead). Images were observed at a magnification of 60,000×. Scale bar, 200 nm. Data are presented as mean ± SEM (n = 3 per group). ** *p* < 0.001 vs. control group; # *p* < 0.001 vs. 4-AP-treated group.

**Figure 8 ijms-23-08752-f008:**
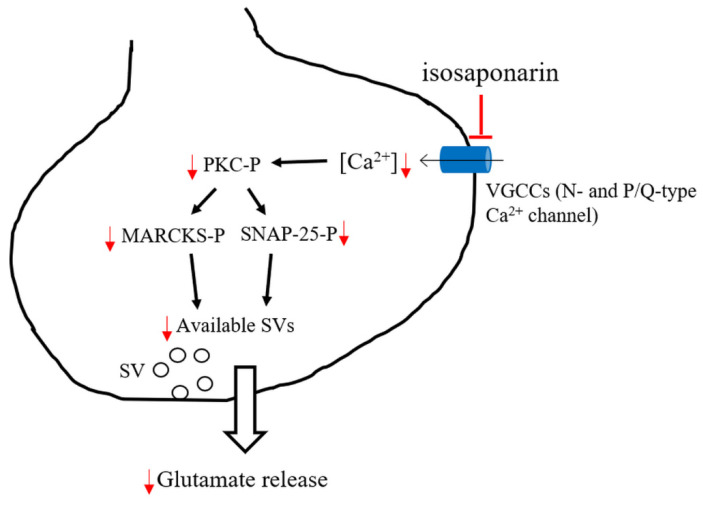
Schematic representation of the pathways involved in the inhibition by isosaponarin of evoked glutamate release from synaptosomes. PKC-P, protein kinase C phosphorylation; SNAP-25-P, synaptosomal-associated protein of 25 kDa phosphorylation; MARCKS-P, myristoylated alanine-rich C-kinase substrate phosphorylation; SV, synaptic vesicle.

## Data Availability

The data presented in this study are available on request from the corresponding author.

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
