# Peer review of "The Effect of Isosaponarin Derived from Wasabi Leaves on Glutamate Release in Rat Synaptosomes and Its Underlying Mechanism"

_ijms, 2022, doi:10.3390/ijms23158752_

Round 1
Reviewer 1 Report
The topic of the manuscript entitled "The effect of isosaponarin derived from wasabi leaves on glutamate release in rat synaptosomes and its underlying mechanism"is interesting, and the manuscript is well designed and organized. But the mnuscript needs revision before publication.
1. Please check the reference format carefully;
2. Others are marked in the text.

Author Response
We thank the reviewer for the critical comments and constructive suggestions.
The topic of the manuscript entitled "The effect of isosaponarin derived from wasabi leaves on glutamate release in rat synaptosomes and its underlying mechanism" is interesting, and the manuscript is well designed and organized. But the mnuscript needs revision before publication.
- Please check the reference format carefully.
As suggestion by the reviewer, the reference format is revised (Page 14-17).
- Others are marked in the text.
The words marked by the reviewer is revised (Page 12, line 362,364; Page 13, line 398; Page 14-17).

Reviewer 2 Report
Abstract:
Comment: The abstract is missing an introductory sentence explaining the importance of the work.
Introduction:
Comment: I suggest replacing the term “antioxidative” with “antioxidant” throughout the manuscript.
Question: Any information about the toxicity of isosaponarin?
Lines 69-70: in vivo and in vitro should be in italic. Here and elsewhere.
Lines 71-76: I do not understand why the outcome of the study is being reported in the introduction. This should be under the results section.
Material and Methods:
Comment: I do not see the details regarding the application of isosaponarin on the synaptosomal samples.
Author Response
We thank the reviewer for the critical comments and constructive suggestions.
Abstract:
Comment: The abstract is missing an introductory sentence explaining the importance of the work.
As suggestion by the reviewer, the sentences〝Excessive glutamate release is known to be involved in the pathogenesis of neurological diseases, and suppression of glutamate release from nerve terminals is considered to be a treatment strategy.〞are added in the abstract (Page 1, Lines 24-26).
Introduction:
Comment: I suggest replacing the term “antioxidative” with “antioxidant” throughout the manuscript.
The word is change to antioxidant (Page 2, line 62).
Question: Any information about the toxicity of isosaponarin?
The information about the toxicity of isosaponarin has not yet been reported.
Lines 69-70: in vivo and in vitro should be in italic. Here and elsewhere
The word is revised (Page 2, line 74).
Lines 71-76: I do not understand why the outcome of the study is being reported in the introduction. This should be under the results section.
As suggestion by the reviewer, the sentences are deleted.
Material and Methods:
Comment: I do not see the details regarding the application of isosaponarin on the synaptosomal samples.
As suggestion by the reviewer, the sentence〝Isosaponarin dissolved in 0.1% dimethylsulfoxide (DMSO) was added 10 min before 4-AP addition and other drugs (bafilomycin A1, ω-conotoxin GVIA, ω-agatoxin IVA, GF109203X, Go6976, and rottlerin) were added 10-20 min before this.〞(Page 13, Lines 444-446).

Reviewer 3 Report
In this manuscript entitled: “The effect of isosaponarin derived from wasabi leaves on gluta‐mate release in rat synaptosomes and its underlying mechanism”.
However, before publication this paper would need revision.
1. Abstract part of manuscript detailed material and methods are missed. Abstracts should be revised and amended with the essential information about the novelty of the work
2. The introduction of the isosaponarin derived from wasabi leavesis somewhat ambiguous and has not fluently and constantly expressed. Therefore, it is needed to be relevant with adequate citations.
3. What is the reason for choosing bioactive compound, isosaponarin?.
4. Scale bar and magnification must be provided for all the images.
5. In western blot experiments, authors must provide adequate details for anti-bodies dilution and sources of all the materials.
Author Response
We thank the reviewer for the critical comments and constructive suggestions.
In this manuscript entitled: “The effect of isosaponarin derived from wasabi leaves on gluta‐mate release in rat synaptosomes and its underlying mechanism”.
However, before publication this paper would need revision.
- Abstract part of manuscript detailed material and methods are missed. Abstracts should be revised and amended with the essential information about the novelty of the work
As suggestion by the reviewer, the sentences〝Excessive glutamate release is known to be involved in the pathogenesis of neurological diseases, and suppression of glutamate release from nerve terminals is considered to be a treatment strategy. In this study, we investigated whether isosaponarin, a flavone glycoside isolated from wasabi leaves, could affect glutamate release in rat cerebral cortex nerve terminals (synaptosomes). The release of glutamate was evoked by the K+ channel blocker 4-aminopyridine (4-AP) and measured by an on-line enzyme-coupled fluorimetric assay.〞are added in the abstract (Page 1, Lines 24-29).
- The introduction of the isosaponarin derived from wasabi leavesis somewhat ambiguous and has not fluently and constantly expressed. Therefore, it is needed to be relevant with adequate citations.
As suggestion by the reviewer, in the abstract section, the sentences〝Numerous flavonoids with antioxidative activity are known to reduce glutamate release and counteract glutamate-induced oxidative damage to neurons [19-21]. In the present study, isosaponarin, a 4′-O-glucosyl-6-C-glucosyl apigenin (Figure 1A), was chosen be-cause of it is one of the flavonoids in wasabi leaves [11,22], but its role in the regulation of glutamate release has not yet been clarified.〞are revised (Page 2, Lines 65-69) and references are citated.
- What is the reason for choosing bioactive compound, isosaponarin?.
About this point, the sentence is modified to〝Numerous flavonoids with antioxidative activity are known to reduce glutamate release and counteract glutamate-induced oxidative damage to neurons [19-21]. In the present study, isosaponarin, a 4′-O-glucosyl-6-C-glucosyl apigenin (Figure 1A), was chosen be-cause of it is one of the flavonoids in wasabi leaves [11,22], but its role in the regulation of glutamate release has not yet been clarified.〞(Page 2, Lines 65-69).
- Scale bar and magnification must be provided for all the images.
In the Figure 6 legend, the sentence〝Images were observed at a magnification of 100×. Scale bar, 30 mm. 〞is added. (Page 8, line 219-220). In the Figure 7 legend, the sentence〝Images were observed at a magnification of 60000×. Scale bar, 200 nm. 〞is added. (Page 9, line 235-236).
- In western blot experiments, authors must provide adequate details for anti-bodies dilution and sources of all the materials.
About this point, the sentences〝The primary antibodies used were PKC (1:600, Abcam, Cambridge, UK), pPKC (1:1000; Cell Signaling, Beverly, MA, USA), PKCa (1:600; Cell Signaling, Beverly, MA, USA), pPKCa (1:2000; Abcam, Cambridge, UK), SNAP-25 (1:20000; Abcam, Cambridge, UK), pSNAP-25 (Ser187) (1:2000; Abcam, Cambridge, UK), pMARCKS (Ser152/156) (1:250; Cell Signaling, Beverly, MA, USA), and b-actin (1:1000; Cell Signaling, Beverly, MA, USA).〞are added in the method section (Page 13, Lines 400-405).

Round 2
Reviewer 3 Report
The revised version of manuscript is acceptable for publication.